# Probing the Parameter Space of Axion-Like Particles Using Simulation-Based Inference

P. Bhattacharjee[1⋆], C. Eckner[1†], G. Zaharijas[1‡], G. Kluge[2§] and G. D'Amico[3¶]
on behalf of CTAO Collaboration

**1** University of Nova Gorica, Slovenia
**2** University of Oslo, Norway
**3** Institut de Física d'Altes Energies (IFAE), Spain

⋆ pooja.bhattacharjee@ung.si † christopher.eckner@ung.si ‡ gabrijela.zaharijas@ung.si §
g.w.kluge@fys.uio.no ¶ gdamico@ifae.es

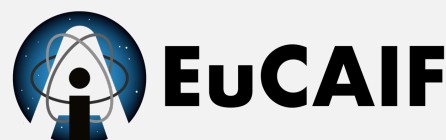

*The 2nd European AI for Fundamental Physics Conference (EuCAIFCon2025) Cagliari, Sardinia, 16-20 June 2025*

## Abstract

Axion-like particles (ALPs), hypothetical pseudoscalar particles that couple to photons, are among the most actively investigated candidates for new physics beyond the Standard Model. Their interaction with gamma rays in presence of astrophysical magnetic fields can leave characteristic, energy-dependent modulations in observed spectra. Capturing such subtle features requires precise statistical inference, but standard likelihood-based methods often fall short when faced with complex models, large number of nuisance parameters and limited analytical tractability. In this work, we investigate the application of simulation-based inference (SBI), specifically Truncated Marginal Neural Ratio Estimation (TMNRE), to constrain ALP parameters using simulated observations from the upcoming Cherenkov Telescope Array Observatory (CTAO). We model the gamma-ray emission from the active galactic nucleus NGC 1275, accounting for photon–ALP mixing, extragalactic background light (EBL) absorption, and the full CTAO instrument response. Leveraging the `Swyft` framework, we infer posteriors for the ALP mass and coupling strength and demonstrate its potential to extract meaningful constraints on ALPs from future real gamma-ray data with CTAO.

# 1   Introduction

Axion-like particles (ALPs) are hypothetical pseudoscalars arising in many Standard Model extensions [1]. Unlike the QCD axion, their mass ($m_a$) and photon coupling ($g_{a\gamma}$) are independent, spanning a broad, unexplored parameter space. Their weak interactions and neutral, spin-zero nature make them compelling dark matter candidates in both cosmology and high-energy astrophysics. In external magnetic fields, ALPs couple to photons via operator $\mathcal{L}a\gamma = -\frac{1}{4}g_{a\gamma}aF_{\mu\nu}\tilde{F}^{\mu\nu}$, where a is the ALP field, $F_{\mu\nu}$ is the electromagnetic field strength, $\tilde{F}^{\mu\nu}$ is its dual, and $g_{a\gamma}$ the coupling constant [2], enabling photon-ALP oscillations. This interaction can imprint energy-dependent features in $\gamma$-ray spectra [3, 4], offering an indirect detection channel with current and future telescopes. We focus on the bright active galactic nuclei (AGN) NGC1275 ($z = 0.0176$) at the center of the Perseus Cluster, whose turbulent magnetic environment ($\sim 10\ \mu$G) provides ideal conditions for photon–ALP conversion in the energy range accessible to the Cherenkov Telescope Array Observatory (CTAO)[5, 6]. The observable flux at detectors, under the influence of ALP mixing, is given by, $\phi_{\text{obs}}(E) = \mathcal{P}_{\gamma\gamma}(E) \cdot \phi_{\text{int}}(E)$, where $\phi_{\text{int}}$ is the intrinsic source spectrum and $\mathcal{P}_{\gamma\gamma}(E)$ is the photon survival probability along the line of sight. Depending on the energy, magnetic field configuration, and ALP parameters, the survival probability may deviate significantly from unity, altering the spectral shape.

Accurate inference of ALP parameters is hindered by significant astrophysical uncertainties, particularly in source modeling, especially related to magnetic field configurations, which introduce numerous nuisance parameters. Traditional likelihood-based methods struggle with high-dimensional marginalization and often rely on strong simplifications. To address this, we adopt Simulation-Based Inference (SBI) using Truncated Marginal Neural Ratio Estimation (TMNRE) [7], a likelihood-free approach suited to complex, nonlinear forward models. This allows us to estimate posteriors for ALP mass, $m_a$, and coupling constant, $g_{a\gamma}$, directly from gamma-ray spectra, bypassing the need for an explicit likelihood and providing a more robust avenue for ALP constraints with CTAO.

# 2   Simulation and Inference Framework

We target the bright AGN NGC 1275 ($z = 0.0176$), located at the center of the Perseus Cluster, where strong, turbulent intra-cluster magnetic fields provide favorable conditions for photon–ALP mixing. Simulated observations are performed using the CTAO *Prod5–North–20deg–AverageAz–4LSTs09MSTs* instrument response functions (IRFs) [8], covering the energy range from ∼10 GeV to 100 TeV. A 50-hour exposure is assumed, representing the observation of a flaring state.

The intrinsic spectrum of NGC 1275 is modeled as an exponentially cut-off power law [6], with amplitude $\Phi$, spectral index $\Gamma$, and cutoff energy $E_{\text{cut}}$ treated as nuisance parameters. For tractability, uncertainties in most of the 15 nuisance parameters are kept fixed. Spectral templates are generated using `gammapy` v1.1 [9], incorporating photon–ALP oscillations and attenuation from the extragalactic background light (EBL) via `gammaALPs` [10]. The

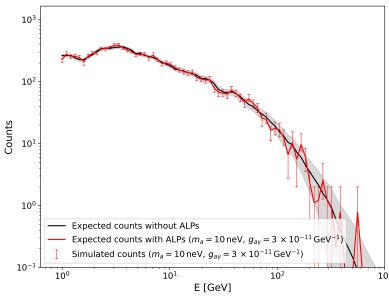

Figure 1: Simulated counts for NGC 1275, showing spectra with ALP-induced oscillations (red) and without (black), together with error bars from simulated counts.

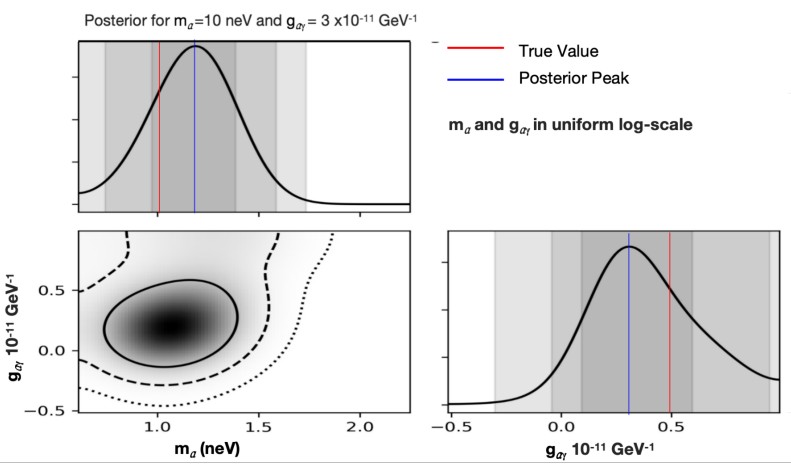

Figure 2: Posterior distribution for simulated true values, $m_a = 10$ neV and $g_{a\gamma} = 3 \times 10^{-11}$ GeV$^{-1}$. The contours peak near the true value but exhibit broad uncertainty due to limited simulation density and parameter degeneracies.

magnetic field is modeled as a Gaussian turbulent field, following [10]. Figure 1 shows the expected counts without ALPs (black) and with ALP-induced modulations for $m_a = 10$ neV and $g_{a\gamma} = 3 \times 10^{-11}$ GeV$^{-1}$ (red), including error bars from simulated counts. These spectral features form the basis of our inference, where we adopt log-uniform priors $m_a \in [1, 10^3]$ neV and $g_{a\gamma} \in [10^{-12}, 10^{-10}]$ GeV$^{-1}$, consistent with refs. [6, 11].

## 3 Results and Calibration

To infer the ALP parameters $(m_a, g_{a\gamma})$, we apply SBI through the SWYFT v0.4.5 framework [12], which leverages neural networks trained on simulated data to approximate likelihood ratios. Training is performed on spectra generated from samples drawn from the joint prior distribution of $m_a$ and $g_{a\gamma}$, using the default marginal classifier architecture and the convergence is monitored via validation loss. While most nuisance parameters are held fixed to reduce computational complexity, variations in the magnetic field are implicitly included, as each simulation uses a randomly generated realization, capturing the stochastic nature of the environment. We choose injection points in the $(m_a, g_{a\gamma})$ plane that are well within the limits of exclusion bounds expected from CTAO [6], and Figure 2 shows the posterior distribution for an injected benchmark point of $m_a$ and $g_{a\gamma}$. The joint posterior peaks close to the true values, indicating that the trained network effectively learns from simulated spectra and is sensitive to

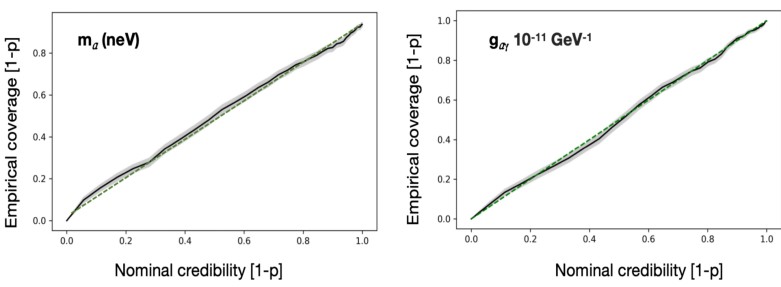

Figure 3: Expected Coverage Probability (ECP) for $m_a$ and $g_{a\gamma}$. While $g_{a\gamma}$ shows good calibration, $m_a$ is slightly underconfident at low credibility.

ALP-induced spectral modulations. However, the contours remain relatively broad, reflecting the limited training sample density and potential degeneracies between parameters.

We assess the calibration of our posterior estimator using the Expected Coverage Probability (ECP) test [13], which compares the empirical coverage of credibility regions to their nominal credibility. As shown in Figure 3, the posterior over $g_{a\gamma}$ tracks the diagonal well (ideally, the empirical coverage should follow the diagonal (green dashed line)), hence not indicating that the inferences are over-or underconfident, on average, at this point in the investigation. In contrast, the $m_a$ marginal undercover at low credibility levels, suggesting mild overconfidence or reduced calibration in that regime.

## 4  Discussion

This work demonstrates the potential of SBI for constraining ALP parameters from future gamma-ray observations with CTAO, as previously explored in Ref. [11]. Using TMNRE, we show that posteriors peak near injected values, confirming sensitivity to ALP-induced spectral modulations. However, broad contours and calibration issues, particularly at low credibility for certain mass ranges, highlight the need for methodological refinement. Future efforts will focus on improving inference quality through:

- **Full training to convergence** with expanded datasets and denser sampling.

- **Incorporation of astrophysical systematics**, including magnetic field and plasma variations.

- **Neural network upgrades**, exploring deeper architectures, alternative loss functions, and calibration techniques such as temperature scaling or conformal prediction.

- **Robustness testing** across different AGN states, spectral shapes, and additional sources.

These steps are critical before applying the framework to real data (e.g., from the Large-Sized Telescopes). With improved calibration, broader parameter coverage, and more realistic systematics, SBI holds strong promise for future ALP searches with CTAO.

## Acknowledgments

We thank the CTAO Consortium and the `gammapy` and the `gammaALPs` developer teams.

**Funding information**    PB acknowledges support from the COFUND action of Horizon Europe's Marie Sklodowska-Curie Actions research programme, Grant Agreement 101081355 (SMASH).

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
