# Peer review of "Probing the Parameter Space of Axion-Like Particles Using Simulation-Based Inference"

_SciPost Physics Proceedings_

## Round 1 · Referee Report · Anonymous (Referee 1) · 2025-12-2

Report

Dear authors, editors,

thanks a lot for these nicely prepared proceedings. I found them well written and clear, so I recommend them to be published with the following minor clarifications: * could you write how many of the nuisance parameters are not kept fixed? Currently you write “most” are kept fixed, but knowing how many aren’t is an important figure * could you clarify what is meant in section 3 with “variations in the magnetic field are implicitly included, as each simulation uses a randomly generated realization, capturing the stochastic nature of the environment”? Are these variations nuisance parameters or you are referring to the fact that the model is an stochastic one, and therefore your studies are really SBI and not a simple regression of the likelihood? Cheers, your referee

Recommendation

Ask for minor revision

---

## Round 1 · Referee Report · Anonymous (Referee 2) · 2025-12-15

Strengths

1) Constraining axion properties with gamma-ray spectra using simulation-based inference (SBI) is a promising idea 2) Their method produces (nearly) unbiased posteriors 3) The paper is well written

Weaknesses

1) The fact that the posteriors are already quite broad - despite the fact that the authors currently keep most of the 15 physical parameters fixed - raises doubts on the ability of the method to obtain strong bounds on axion properties in a more realistic setting where all physically relevant parameters need to be inferred or marginalized over 2) The plot labels could be improved

Report

The authors present an SBI-based analysis of Cherenkov Telescope Array Observatory (CTAO) data - specifically of an AGN at the center of the Perseus Cluster - aiming to constrain axion-like particles (ALPs). The idea of the paper is interesting, and the authors show that their method produces roughly unbiased, albeit quite broad, constraints. While I think that, based on these present proof-of-concept results, it remains to be seen how strongly ALP properties can be constrained with this approach, this submission presents a valuable stepping stone toward a more comprehensive SBI-based analysis framework. Therefore, I recommend the paper for publication, but I kindly ask the authors to first address my comments below.

Requested changes

1) In the operator $\mathcal{L}_{\alpha \gamma}$, $\alpha$ and $\gamma$ should be subscripts 2) "For tractability, uncertainties in most of the 15 nuisance parameters are kept fixed" -> The clarity of this sentence could be improved, as it's not entirely clear at this point if the values of the nuisance parameters themselves or only their standard deviations are kept fixed (while their values would be allowed to float) 3) The font size in the legend, as well as of the tick labels, of Fig. 1 is very small - please consider increasing it (there's still empty space in the figure) 4) The labels of the posterior results in Fig. 2 are somewhat misleading: the x-axis of the $m_a$ plot reads $m_a$, but what seems to be shown is actually $\log_{10} m_a$, and similarly for the coupling strength. Please make the axis labels consistent with the data / values indicated by the tick labels. 5) "...which leverages neural networks trained on simulated data to approximate likelihood ratios." -> "Likelihood ratios" is very generic; it would be good to mention that swyft specifically targets the posterior-to-prior or equivalently the likelihood-to-evidence ratio 6) "While most nuisance parameters are held fixed to reduce computational complexity, variations in the magnetic field are implicitly included, as each simulation uses a randomly generated realization, capturing the stochastic nature of the environment. " -> the authors seem to infer 2 parameters (mass and coupling strength) and to marginalize over another parameter (magnetic field). Can the authors please state this clearly already at an earlier point where they wrote "For tractability, uncertainties in most of the 15 nuisance parameters are kept fixed." 7) Please mention the input dimensionality (i.e. number of energy bins) of the spectra shown to the neural network. 8) The authors write: "However, the contours remain relatively broad, reflecting the limited training sample density and potential degeneracies between parameters." It would be good to mention the size of your training dataset and perhaps also to comment on what makes you believe that increasing the training dataset size would lead to much tighter constraints. 9) The future effort enumeration lists "Incorporation of astrophysical systematics, including magnetic field..." -> but earlier you wrote: " variations in the magnetic field are implicitly included, as each simulation uses a randomly generated realization"; please clarify to what extent this is already included and what is still missing 10) The authors seem to be worried about the calibration, but in my opinion, their coverage plot looks decent. What worries me more is that the constraints in Fig. 2 look quite broad (as the authors note themselves) - despite the fact that most of the 15 nuisance parameters mentioned by the authors are kept fixed. On the other hand, this work only considers a single AGN. Could the authors briefly discuss the potential of their method for deriving ALP constraints jointly from multiple sources?

Recommendation

Ask for minor revision

---

## Editorial Decision

awaiting_resubmission